# Numerical and Experiment Investigation on Novel Guide Vane Structures of Turbo Air Classifier

**Yun Zeng** , **Bowen Huang, Daoxin Qin, Sizhu Zhou and Meiqiu Li** *

School of Mechanical Engineering, Yangtze University, Jingzhou 434023, China; mechanicszy@163.com (Y.Z.); huangbw_asdf@163.com (B.H.); q_heguang@163.com (D.Q.); zhousz_fracturing@163.com (S.Z.)
* Correspondence: limq@yangtzeu.edu.cn

**Abstract:** In this paper, three types of air guide vanes are designed: direct-type, L-type, and logarithmic spiral type, respectively. ANSYS-FLUENT 20.0 is used to numerically simulate the internal flow field of turbo air classifier by novel different structures. The numerical results show that the guide vane structures have a good effect on the flow field stability of the annular function zone in the classifying chamber. The distribution of tangential velocity and radial velocity verified the logarithmic spiral guide vane, and makes the airflow flow along the rotor cage circumferentially uniformly. In addition, the turbulent dissipation rate and energy loss decreases in the rotor cage region, which also shows that the guide vane is beneficial to improve classification performance. The tromp curve of the numerical simulation shows that the logarithmic spiral guide vane reduced the cutting size by 6.3% and 23.7% at two different process parameters, and is obviously better than other guide vane structures in improving the classification sharpness index (K). Finally, the reliability of numerical simulation is verified by material experiment. The research results have certain theoretical significance and guidance for the structural design of the guide vanes of the turbo air classifier.

**Keywords:** turbo air classifier; tromp curves; numerical simulation; guide vane; particle classification efficiency

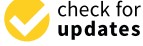



## 1. Introduction

With the rapid development of the powder preparation industry, Ultrafine powder has been widely used in a variety of different fields. It is a promising material in the field of new materials research, which is full of vitality and has a very important influence on future economic and social development. The traditional processing method of ultrafine powder is mainly to crack and crush the coarse material with mechanical force. Commonly used equipment includes planet grinding, airflow grinding, and so on. However, in the grinding process, only part of the powder satisfies the particle size requirements. If the powder that meets the particle size is not separated out in time, the result is crushed with other powder again, wasting resources. Therefore, powder classification equipment is very important in practical production. Turbo air classifier, as one of the core pieces of equipment for producing ultrafine iron ore powder, is widely used in chemical, petroleum, pharmaceutical, and other industries [1–3]. However, industrial circles have higher requirements for ultrafine powder with the development of technology. Consequently, it is difficult for the turbo air classifier to satisfy the production requirements. In order to improve the performance of the turbo air classifier, many scholars [4,5] have investigated the effect of process parameters on classification performance. Nevertheless, these improvements only satisfied the requirement of specified samples production. It is difficult to satisfy the classification performance of different particle size distribution and different materials by one kind of classification process parameter. Therefore, it is not enough to improve classification performance only from process parameters. Instead, scholars choose to improve and optimize the structure of turbo air classifier for improvement powder preparation.

In general, there are three important grading functional zones in the classifying chamber: the coarse classification zone, annular classification zone and blade classification zone, respectively. Figure 1 shows this in detail. Firstly, the coarse classification zone is the periphery area of the classifying chamber. This region is located between the outer edge of the guide vanes and the internal face of the classifying chamber. Most of the coarse powder in this area received a centrifugal force that is greater than the drag force, and eventually powder that has hit the cylinder wall is accumulated. In addition, the other two grading functional regions are major classification zones. The annular classification zone surrounds the inner edges of the guide vanes and outer edges of the turbo blades, which are called the decentralized separation functional zone. Due to the difference of the forces on the particles in the region, this is the main functional region for classifying the coarse powder and fine powder. Finally, the blade classification zone is the flow region between the blades, which has a certain function of conveying fine powder. Simultaneously, the vortex phenomenon in the rotor cage is one of the factors leading to the decline of classification performance. Therefore, the flow field moving pattern of three classification zones has become an important research factor.

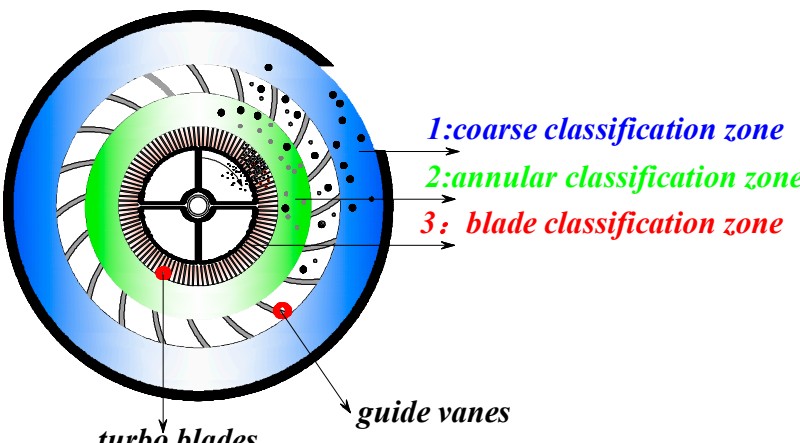

**Figure 1.** Schematic diagram of grading functional area of turbo air classifier.

Many studies [6–10] have been conducted on improving the structure of different parts in the turbo air classifier. Ren [11] designed a rotor cage with non-radial arc blades which reduced the cut size by 11.5% and increased the classification accuracy. Zhao [12] studied the rotor cage chassis with both open and closed structures, the results indicated that the open rotary cage chassis contributes to the uniform distribution of flow field moving patterns. Both classification accuracy and classification sharpness index have been improved. Wu [13] proposed a new double layer spreading plate based on theoretical analysis on the material dispersion. This was able to improve the material dispersion and reduce the probability of collision and aggregation of particles in a turbo air classifier. The experimental results indicated that classification performance is improved. From the above research, it can be found that guide vanes and rotor cage blades are the main research objects. In addition, due to the flow pattern is the main factor affecting the flow field. Sun [14–17] proposed a new horizontal turbo air classifier equipped with two inclined air inlets. It provides a new structure improvement direction based on the structure of the turbo air classifier. In conclusion, the main factor to improve the classification performance is to enhance the flow field in the classifying chamber. Due to the lack of theoretical guidance and the cost of processing the guide vane in practical engineering, there is little research on guide blades. In this paper, the flow trajectory in the internal flow field of the classifier is investigated theoretically, and the equation of the air flow trajectory is deduced. The authors designed three types of guide blades based on the air flow trajectory equation, and the influence of different structure of air guide blades on the internal flow of the classifier

was principally inspected to provide theoretical guidance for the following optimization design of the turbo air classifier.

## 2. Description of the Equipment

### 2.1. Overall System

A scheme diagram of the pneumatic screening system and turbo air classifier is shown in Figure 2. The initial grinding of the material is carried out between the original grinding roller and the grinding ring, and then grinding is undertaken four times. Therefore, the material can be completely ground, and we obtain a certain fine particle size product. Nonetheless, some of the fine-grained products still do not meet the requirements of iron powder in the drilling and completion fluid formulation, so they need to be screened by a classifier. The whole pneumatic screen system is divided into primary classification, secondary classification and pulse dust removal system, respectively. This paper only studies on the first stage classifier.

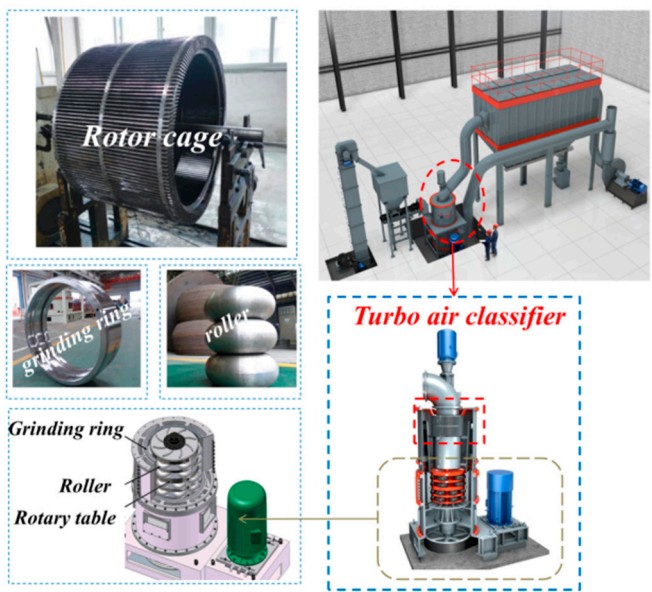

**Figure 2.** Scheme diagram of the pneumatic screening system.

### 2.2. Working Principle

As aforementioned, various studies [18,19] have verified that the guide vane has an obvious effect on the flow field of annular region in turbo air classifier. More critically, the numerical value of flow field velocity and the uniformity of velocity distribution in the annular region play a decisive role in the classification performance of coarse and fine particles. According to the literature [20,21], the air streamline inside the classifying chamber of the classifier shows a law of spiraling upward movement. The corresponding schematic diagram is shown in Figure 3, and structural dimensions of the turbo air classifier are shown in Figure 4.

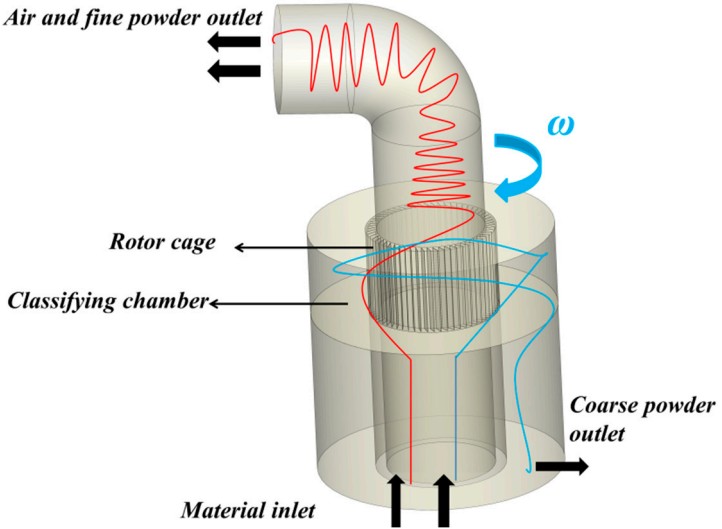

**Figure 3.** Working principle of the turbo air classifier.

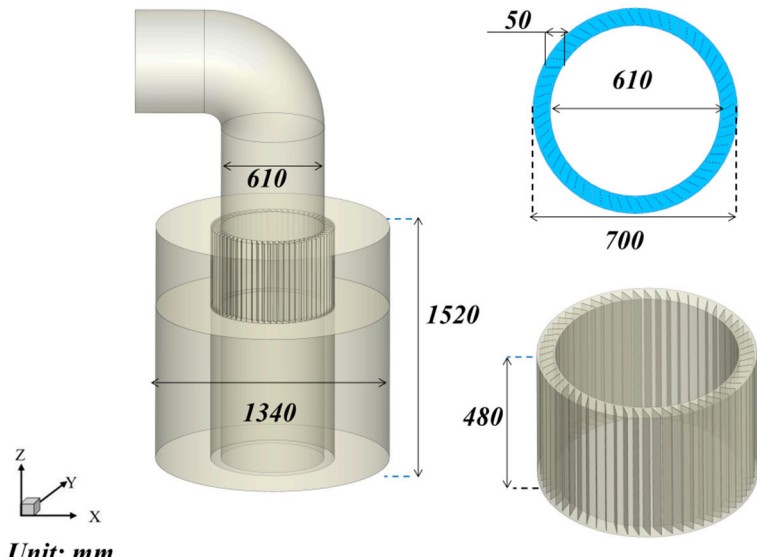

Unit: mm

**Figure 4.** The structural dimensions of the turbo air classifier.

## 3. Details of the Calculation Methodology

### 3.1. Derivation of Shape Design Formula of Guide Vane

In order to further simplify the study, the air flow streamline of turbo air classifier at the annular region is described by geometric relation, and the motion of airflow is shown in Figure 5. The dotted line is the outer edge of the rotor blades, and thick solid line BC is the trajectories of fluid particles. $\varphi$ is the tongue angle between the airflow section and line OC. It is defined as airflow cross section angle.

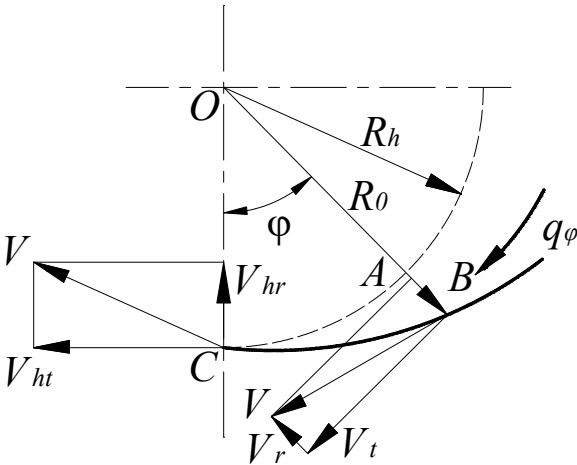

**Figure 5.** Analyses of the motion of airflow in the volute.

According to the geometric relationship of airflow movement illustrated in Figure 5, it can be assumed the air rate of flow into section AB as $q_\varphi$. This part of air flow will enter the annular classification region from the outer edge of the rotor blades arc AC, due to the tongue angle between the two guide vanes is $\varphi$, $R_0$ is the distance from point on center line of guide vane to center of rotor cage, the following equation about tongue angle and air rate of flow into section can be derived as:

$$q_\varphi = \frac{Q}{2\pi}\varphi \tag{1}$$

Ignoring the influence of friction, the air flow inside the classifying chamber is not affected by the external force, it meets the conservation law of moment of momentum. The velocity of fluid element relation can be expressed as follows:

$$R_h V_{ht} = R_0 V_t = C(\text{constant}) \tag{2}$$

$V_{ht}$ and $V_t$ are the tangential velocity of airflow fluid element with radius $R_h$ and $R_0$.

The section AB is all effective airflow, and the flow rate $q_\varphi$ of the section can be derived as:

$$q_\varphi = \int_{R_h}^{R_0} hV_t dR \tag{3}$$

$h$ is the height of the runner in blade classification region, which is 0.740 m in this paper, in order to unify analysis, the paper guide vanes height is according to this design. Combined with Formulas (3) and (4), it can be deduced as follows:

$$q_\varphi = \int_{R_h}^{R_0} h\frac{R_h V_{ht}}{R_0}dR = hR_h V_{ht} \ln\frac{R_0}{R_h} \tag{4}$$

Finally, the following equation can be derived from Equations (1)–(4):

$$R_0 = R_h e^{\frac{Q}{2\pi h R_h V_{ht}}\frac{\pi}{180}\varphi}, V_{ht} = \frac{n\pi R_h}{30} \tag{5}$$

### 3.2. Design of Different Guide Vane Shapes

Due to the trajectory of the air flow from the interior wall of the classifying chamber to the outer edge of the guide vane is a logarithmic spiral. The proportion of powder in the classifying chamber is in the dilute phase, which belongs to gas-solid two-phase flow. It can be assumed that the particle movement path mainly moves with the airflow. Therefore, the flow law of gas in the classifying chamber has a particularly important effect on the classification performance. Until now, various studies [22,23] have verified

the shape of blade effect on the law of flow field movement in turbo fluid machinery. In particular, the influence of the inlet effect will cause layer separation as the airflow enters the rotating mechanical blade. Therefore, in order to reduce the influence of import effect on classification performance, a new type of guide vane was proposed based on the theory of Section 3.1. The novel designed guide vane is defined as the logarithmic spiral guide vane. Moreover, another two different types of guide vane were similarly designed to compare with the original logarithmic spiral guide vane. In order to further design the size of the guide vane, the size of the rotor cage is a necessary reference parameter. In this paper, the specific parameters of the rotor cage blade are as follows: length 50 mm, width 4 mm, height 480 mm, inlet mounting angle 60°. Moreover, from the existing literature [24], the ratio between the width $D_1$ of the annular classification functional region and the diameter of the rotor blade *2r* can be 0.09–0.12, and we refer to the current widely used classifier and industry standards for the following design:

(1)  The parameters of the direct-type guide vans are: 55 mm in length, 4 mm in width and 480 mm in height.
(2)  The groove surface of L-type air guide vane faces the runner blade, and the parameters of three sections of blade are as follows: The width and height of the three blades are uniform, the length of numbers 1, 2 and 3 are 60 mm, 25 mm, 25 mm, respectively. According to the runner blade installation angle, $\varphi_1$ and $\varphi_2$ are 30° and 60° respectively.
(3)  Design of logarithmic helix air guide blade. According to Formula (5), streamline (particle track) of airflow particle is like logarithmic helix. Therefore, the radius of logarithmic spiral air guide blade is calculated to be 0.729 mm, the length of it is close to the direct-type guide vane.

Figure 6 shows the schematic diagram of the three types of air guide blades, and the specific parameters are shown in Table 1.

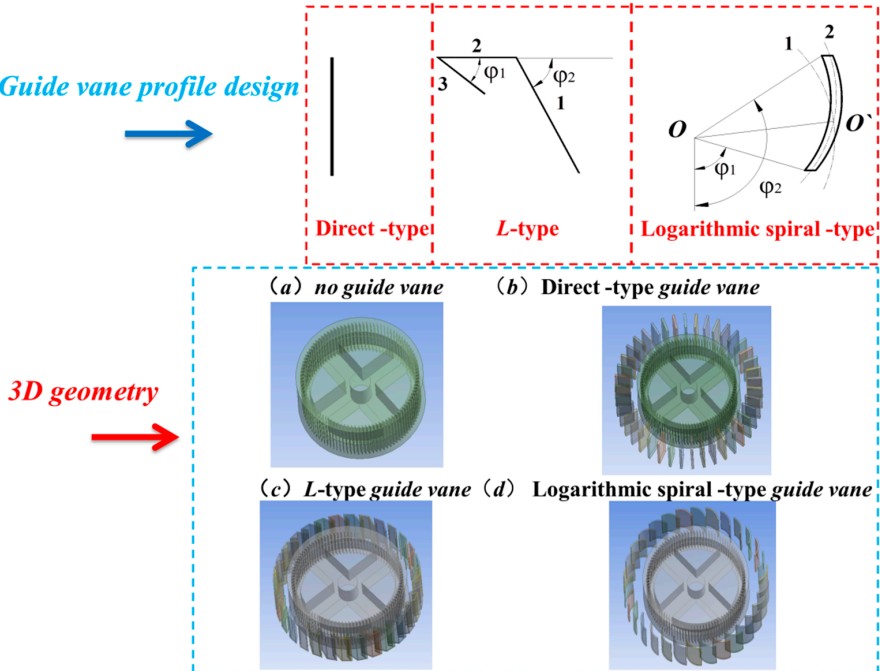

**Figure 6.** Different types of guide vanes design.

**Table 1.** Design of three different blade structural parameters.

| | Numbers of Vans | $\varphi_1, \varphi_2$ | Long/Arc Radius | | | Width | Height |
|---|---|---|---|---|---|---|---|
| Direct-type | 40 | 0° | | 75 mm | | 4 mm | 480 mm |
| L-type | 30 | 30°, 60° | 1 60 mm | 2 25 mm | 3 25 mm | 4 mm | 480 mm |
| Logarithmic spiral-type | 30 | 89°, 92° | | R = 729 mm | | 4 mm | 480 mm |

*3.3. Numerical Calculation Model and Simulation Conditions*

3.3.1. Turbulence Model Description

Many researchers [5,19,20] has verified that the flow field in turbo air classifier is turbulent flow and swirl flow field, and the standard $k$-$\varepsilon$ model is not suitable for simulating flow conditions near curved walls. In contrast, (RNG $k$-$\varepsilon$) and Reynolds Stress Model (RSM) can both be suitable for simulating the flow field of the turbo air classifier. However, the computational cost required by the RSM model is too high, and the result is difficult to converge; it is not suitable for engineering applications. Hence, the (RNG $k$-$\varepsilon$) model can be selected as the suitable turbulence model for calculating the flow field of the turbo air classifier. For the case of incompressible flow, the mass and momentum equations are as follows:

$$\frac{\partial u_i}{\partial x_i} = 0 \tag{6}$$

$$\frac{\partial}{\partial t}(\rho u_i) + \frac{\partial}{\partial x_j}(\rho u_i u_j) = -\frac{\partial p}{\partial x_i} + \frac{\partial}{\partial x_j}(\mu \frac{\partial u_i}{\partial x_j} - \rho \overline{u'_i u'_j}) + S_i \tag{7}$$

where $u_i$, $x_i$, $\rho$, $p$, and $\mu$ represent the fluid velocity, position time, constant fluid density, static pressure, and gas viscosity, respectively. $-\rho \overline{u'_i u'_j}$ is the Reynolds stress term. $S_i$ is the interaction force between the continuous phase and the discrete phase, generally regarded as the mass force of the particle group per unit volume acting on the gas. In this paper, the influence of particles on the flow field is ignored. However, the Reynolds stress term $-\rho \overline{u'_i u'_j}$, which is a second order correlation term of pulsating velocity, means that the basic governing equations cannot be closed. Therefore, different turbulence models can be formed by using different treatment methods for $-\rho \overline{u'_i u'_j}$. For the RNG $k$-$\varepsilon$ model, the turbulent kinetic energy and turbulent dissipation rate are expressed as follows:

$$\rho \frac{dk}{dt} = \frac{\partial}{\partial x_i}\left[(\alpha_k \mu_{eff})\frac{\partial k}{\partial x_i}\right] + G_k + G_b - \rho \varepsilon - Y_M \tag{8}$$

$$\rho \frac{d\varepsilon}{dt} = \frac{\partial}{\partial x_i}\left(\alpha_\varepsilon \mu_{eff}\frac{\partial \varepsilon}{\partial x_i}\right) + C_{1\varepsilon}\frac{\varepsilon}{k}(G_k + C_{3\varepsilon}G_b) - C_{2\varepsilon}\rho\frac{\varepsilon^2}{k} - R \tag{9}$$

where $G_k$ and $G_b$ represent the components of the turbulent kinetic energy caused by the average velocity gradient and buoyancy. $Y_M$ is the effect of compressible turbulent pulsation expansion on the total dissipation rate. The values of the constant are $\alpha_\varepsilon = 0.7692$, $\alpha_k = 1$, $C_{1\varepsilon} = 1.44$, $C_{2\varepsilon} = 1.92$, $C_{3\varepsilon} = 0.09$

The turbulent viscosity coefficient can be calculated as:

$$\mu_t = \rho C_u \frac{k^2}{\varepsilon} \tag{10}$$

where $C_u = 0.0845$.

3.3.2. Discrete Phase Model

In this paper, according to the production needs of actual equipment, the volume loading rate of the particle phase is less than 10%, which satisfies the DPM calculation conditions. Some researchers indicated that the force of particles on the gas can be ignored

in the gas-solid two-phase flow. Therefore, unidirectional coupling was used to calculate the gas-solid two-phase flow. The force balance equations for discrete particles can be written as:

$$\frac{du_P}{dt} = F_D(u - u_P) + \frac{g_x(\rho_P - \rho)}{\rho_P} + F_x \tag{11}$$

$$F_D = \frac{18\mu}{\rho_P d_P^2} \frac{C_D Re}{24} \tag{12}$$

$$Re_P = \frac{\rho d_P|u_P - u|}{\mu} \tag{13}$$

where $F_D(u - u_P)$ is the drag force per unit particle mass, $u$ is the fluid phase velocity, $u_P$ is the particle velocity, $\mu$ is the kinematic viscosity of fluids, $\rho$ is the fluid density, $\rho_P$ is the particle density, $d_P$ is the particle diameter, $Re$ is the relative Reynolds number (particle Reynolds number) (the define of Particle Reynolds number can be calculated by Equation (19)) $C_D$ is the drag coefficient, and $F_x$ is an additional acceleration (force/unit particle mass) term.

### 3.3.3. Boundary Conditions

ANSYS-fluent 2020R2 software was used for numerical simulation, combine with the actual conditions, "velocity-inlet" and "pressure-outlet" boundary were used at the air-material inlet and fine powder outlet, respectively. The no-slip boundary condition was used at the wall surface, and the near wall surface is the standard wall function. SIMPLEC algorithm was used for pressure-velocity coupling, and QUICK difference scheme was used for convection and diffusion. The convective term of discrete equation adopts a default format, and the relaxation factor is selected by experience. In total, 2000 steps were iterated, and the solution accuracy was set to $1 \times 10^{-3}$. In order to ensure that the grid has no effect on the numerical simulation, the grid independence was verified, and the grid number was ultimately determined to be 1,902,235.

## 4. Analysis of the Numerical Simulation Results
### 4.1. Effect of the Guide Vanes on Velocity Distribution in Annular Region
#### 4.1.1. Tangential Velocity Distribution

The process parameters of the turbo air classifier studied in this paper are: rotation speed of the rotor wheel 220 rpm, and range of air volume 12,500 m$^3$/h. The velocity distribution of internal flow field is the key factor affecting the classification performance in a turbo air classifier. Two different height rotor cage sections Z = 0.22 mm and Z = −0.22 mm were selected for analysis. Figure 6 shows the tangential velocity distribution in two rotor cage sections by four different guide vane structures. Similarly, in the rotor cage region, the tangential velocity of the four structures increases as the rotor cage radius increases. The reason for this is that the tangential velocity of airflow belongs to forced vortex in the rotor cage; therefore, the tangential velocity of airflow in the four structures increases with the increase of radius under different working conditions. However, in the annular functional region, comparing the non-guide vane with the other three structures, the tangential velocity of air flow decreases rapidly. In contrast, the tangential velocity of air flow is relatively stable in the three annular functional regions which contain guide vanes. In conclusion, it can be clearly seen from Figure 7a,b that the tangential velocity of air flow increases at first, and then decreases in the rotor cage region and the annular region, respectively. This phenomenon is typical of a Rankine vortex. In addition, it can be found that tangential velocity had an approximate linear distribution in the rotor cage region when the rotor cage speed was 220 rpm. As is shown in Figure 6, in the annular functional region, the tangential velocity gradient without guide vanes varies greatly than three guide vane structures. Therefore, it is inferred that the tangential velocity of the air flow in this region has a certain decline under the action of the air guide vanes, which is relatively stable. The influence of the guide vanes could make the tangential velocity of the airflow

more stable, which is conducive to the classification of fine particles, and improves the classification accuracy.

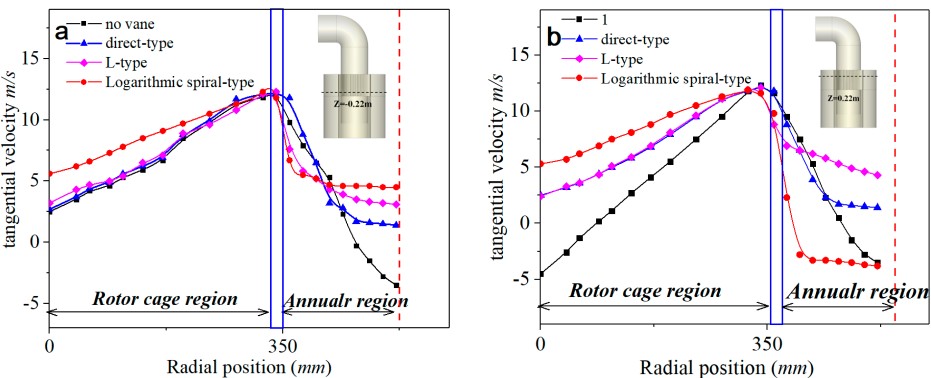

**Figure 7.** Comparison of the tangential velocity distribution of rotor cage by four different guide vanes in two sections (**a**) Z = −0.22 m, (**b**) Z = 0.22 m.

### 4.1.2. Radial Velocity Distribution

Figure 8a,b shows the radial velocity of air flow in the rotor cage region and the annular region, respectively. In the outer edge of the rotor cage and the annular functional region, the guide vanes obviously improved the radial velocity of the airflow in the classifying chamber. Generally, the outer edge of the cage is the interface of the classifying coarse and fine particles during the classification process. The larger radial velocity increases the fluid drag force on the particles, which makes it easier for the coarse particles to enter the rotor cage and be collected into fine powder. Finally, the particle cut size increases. However, in the annular functional region, the radial velocity distribution is uniform under the two guide vane structures L-type and logarithmic-spiral-type, indicating that these two structures are beneficial to the stability of radial velocity distribution in the annular region. It can be inferred that guide vanes contribute to uniform particle size classification, thereby improving the classification sharpness index.

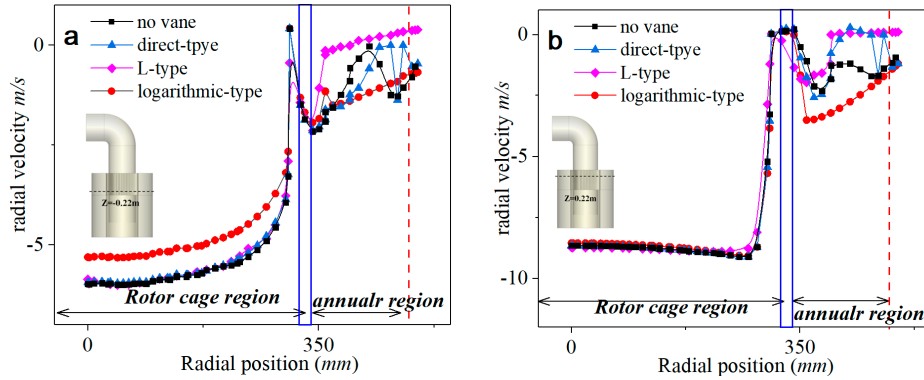

**Figure 8.** Comparison of the radial velocity distribution of rotor cage by four different guide vanes in two sections (**a**) Z = 0–0.22 m, (**b**) Z = 0.22 m.

In order to further analyze the radial velocity uniformity in annular region, for statistical analysis of the radial velocity distribution in the annular functional area, the standard deviation calculation formula is as follows:

$$S = \sqrt{\frac{\sum\limits_{i=1}^{n} \left(X_i - \overline{X}\right)^2}{n-1}} \tag{14}$$

In Z = −0.22 mm section, the standard deviations of radial velocity by four different structures in annular functional zone are *no vane*-0.617 m/s, *direct-type*-0.431 m/s, *L-type*-0.271 m/s, *logarithmic-spiral-type*-0.151 m/s, respectively. In Z = 0.22 mm section, the standard deviations of radial velocity by four different structures in the annular functional zone are *no vane*-0.971 m/s, *direct-type*-0.701 m/s, *L-type*-0.621 m/s, *logarithmic-spiral-type*-0.366 m/s, respectively. It can be concluded that L-shape and logarithmic spiral guide vane structures have certain effects on the improvement of classification performance.

### 4.1.3. Axial Velocity Distribution

Figure 9 shows the axial-velocity distribution on the outer edge surface of the rotor cage. It can clearly be seen that the majority region of the axial velocity on the outer edge surface is close to zero. The values of axial-velocity range from −10 m/s to 10 m/s. It can be inferred that the axial velocity fluctuation occurs at the junction of the rotor cage and the fine particles outlet passage. However, combined with the results of tangential velocity and radial velocity, the axial velocity of the main classifying functional regions fluctuates slightly. The above results indicate that guide vanes have a minor effect on the distribution of axial velocity in classifying chamber. The distribution of axial velocity may be affected by improving other structures in the turbo air classifier.

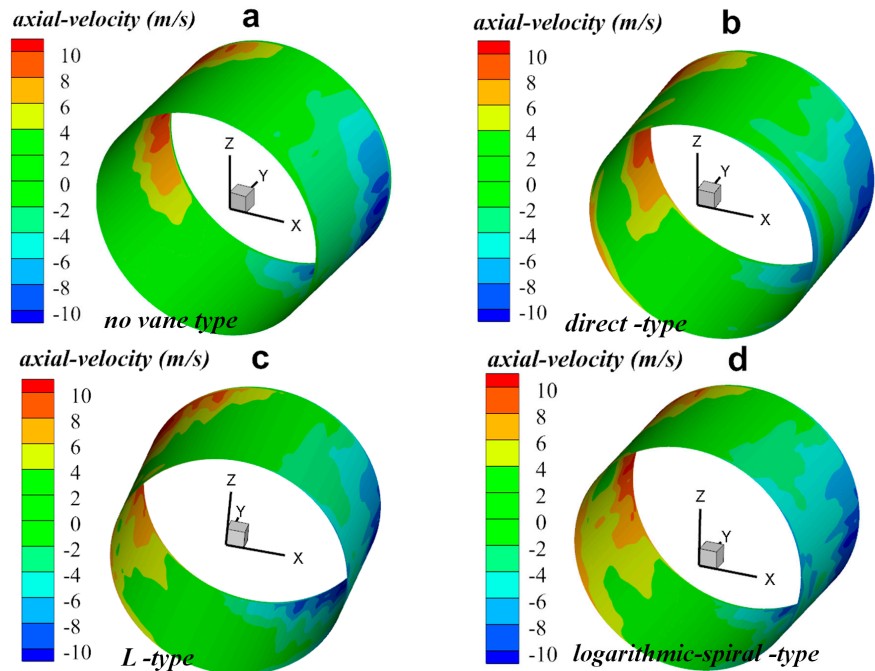

**Figure 9.** Comparison of the axial velocity distribution of rotor cage by four different guide vanes. (**a**) no vane type; (**b**) direct-type; (**c**) L-type; (**d**) logarithmic-spiral-type.

### 4.2. The Effect of the Guide Vanes on the Annular Region Stability

4.2.1. Comparison the Flow Field of Annular Functional Zones for Four Structures

As aforementioned, various studies [18–20] have verified that annular functional region plays an important role in affecting the classification performance in the classifying chamber. As is seen in Figure 10, the Z = 0 mm section was selected for analysis, the flow field in annular classification region is unstable under the two guide vane structures direct-type and no-vane-type, and the velocity gradient varies greatly. Moreover, we compared the velocity distribution in the outer edge region of the guide vanes and the annular functional region, and studied other symmetric structures including L-type guide vanes and logarithmic-spiral-type guide vanes. It was found that the various of velocity gradients for these two structures are close to 0, and the value of the majority velocity magnitude was predominantly at 3 m/s. In particular, the velocity magnitude distribution

is most stable in the regions under the guide vane structures of logarithmic-spiral-type. In conclusion, this indicates that the structure of the guide vanes can significantly improve the stability of the flow field in the classifying chamber, and the effects of L-shaped and logarithmic spiral air guide vanes are better.

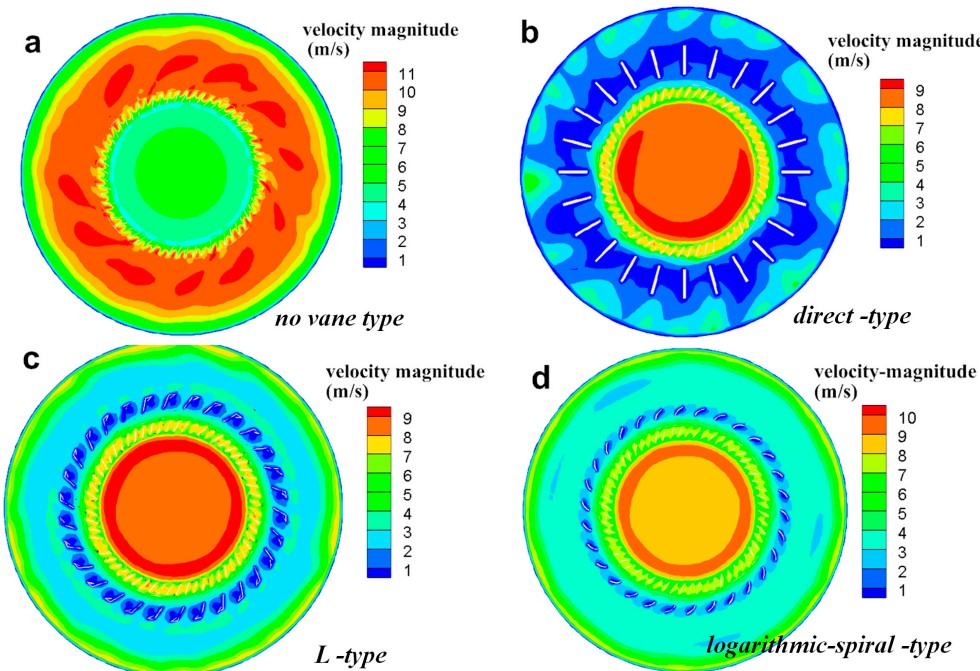

**Figure 10.** Comparison of the velocity magnitude distribution of the rotor cage by four different guide vanes in (Z = 0 m) section, (**a**) no vane type; (**b**) direct-type; (**c**) L-type; (**d**) logarithmic-spiral-type.

### 4.2.2. Comparison of the Turbulent Energy Dissipation by Four Structures

Turbulent dissipation rate can reflect the impact of small-size turbulent vortexes on dispersed materials. The turbulent dissipation rate of the four structures is shown in Figure 11. It is observed that the values of the turbulent dissipation rate value vary in the range of 100–2000 $m^2/s^3$, wherein the smallest value change interval is the structure without guide vanes. However, the turbulent dissipation rate at the inner edge of the rotor cage is larger under the two guide vane structures no-vane-type and direct-type. This indicates that the flow field in the rotor cage region is unstable. According to previous research [4], the rotor cage region and annular region are called the classifying functional zone and decentralized classifying functional zone, respectively. Therefore, the stability of the flow field in the rotor region is critical. In addition, the misplacement offine particles and the bypass phenomenon will significantly affect the classification efficiency. The reason for this phenomenon is the turbulent dissipation rate. Comparison with the turbulent dissipation rate distribution of the four guide vane structures shows that, in the logarithmic spiral guide vane structure, there are few places where the turbulent dissipation rate increases in the rotor cage region. Minor turbulent dissipation rates will result in improved classification efficiency [25–27]. Consequently, guide vane structures L-type and logarithmic-spiral-type could improve the stability of the flow field in the rotor cage region, thereby improving the classification efficiency.

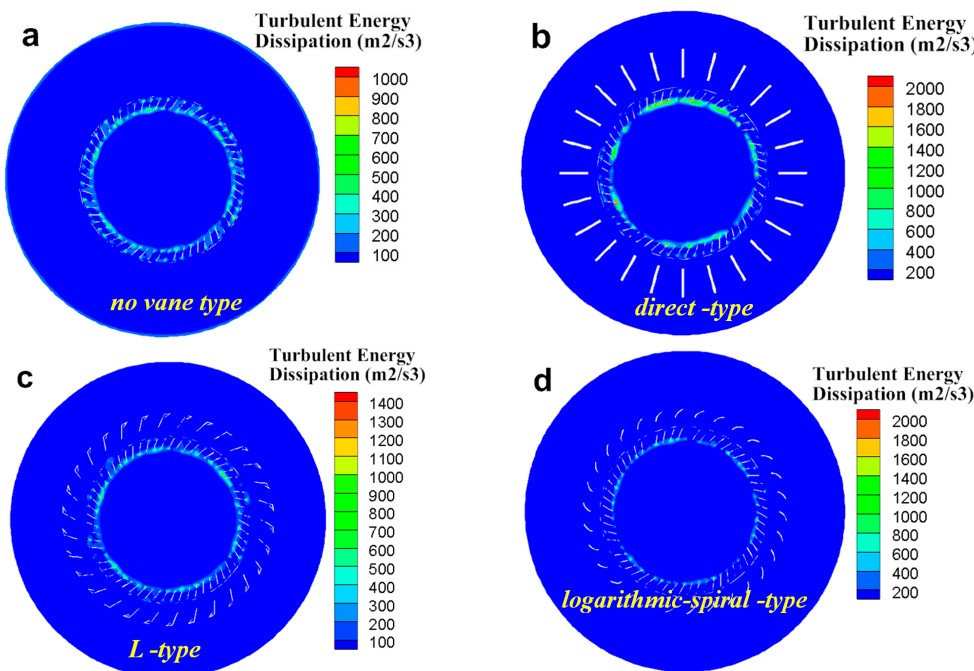

**Figure 11.** Comparison of the turbulent energy dissipation of rotor cage by four different guide vanes in (Z = 0 m) section. (**a**) no vane type; (**b**) direct-type; (**c**) no vane type; (**d**) direct-type.

### 4.2.3. Influence of Different Guide Vanes on Energy Loss

In order to further investigate the classifying mechanism of the classifier, it is indispensable to analyze the energy loss of the flow field in the classifying chamber. Figure 12 shows the static pressure distribution of the Z-section (−0.22 mm and 0.22 mm) in the classifying chamber under four guide vane structures. It can be observed that negative pressure appears in the Z-section (−0.22 mm) of the rotor cage region. However, in contrast to the other three structures, the negative pressure always existed in the rotor cage and annular region of the logarithmic-spiral-type guide vanes, and the fluctuation range of static pressure is smaller than the other three cases. This indicates that energy loss is stable. In addition, in the Z-section (0.22 mm) of the annular region, the static pressure value shows a decreasing trend, indicating that the obstruction of the air flow by the guide vanes will cause the loss of energy.

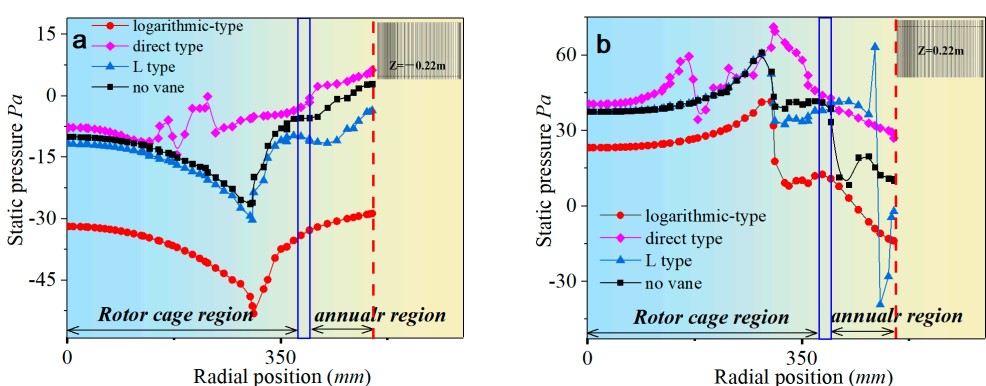

**Figure 12.** Comparison of the static pressure of rotor cage by four different guide vanes in two sections (**a**) Z =−0.22 m, (**b**) Z = 0.22 m.

In Figure 13, the total pressure of the four structures decreased mainly at the outer edge of the rotor cage. This phenomenon is related to the pressure distribution of the Rankine vortex; tangential and radial velocities also affected it. In addition, in the Z-

section (−0.22 mm) of the annular region, the value of tangential velocity is much greater than the radial velocity under logarithmic-spiral-type conditions, and the energy loss is relatively minor.

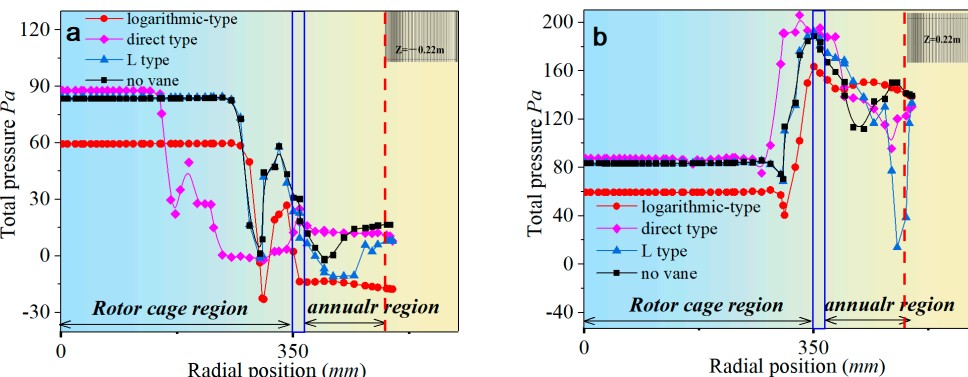

**Figure 13.** Comparison of the total pressure of rotor cage by four different guide vanes in two sections (**a**) Z = −0.22 m, (**b**) Z = 0.22 m.

### 4.3. Effect of the Guide Vanes on the Particle Classification Efficiency

Particle classification efficiency which usually used as an important index to evaluate the particle classification performance, the calculation formula is as follows:

$$\Delta\eta = \frac{w_{coarse}}{w_{material}} \times 100\% \tag{15}$$

where, $w_{coarse}$ and $w_{material}$ are the mass of the collected coarse fraction and the raw material between any particle size range $d$ and $d + \Delta d$.

In order to investigate the particle classification efficiency (or Tromp curve) at different guide vane structures, two process parameters were selected for numerical simulation. Multi-particle discrete phase simulation was carried out, and particles with different sizes released from the same material inlet surface were tracked. The particles escaping from the outlet were captured and collected for statistics. Under the two process parameters, the particle classification efficiency and cut size of the four structures is shown in Figure 14.

According to the results of particle track, the Tromp curve was drawn, and the particle cut size $d_{50}$ was calculated by the Tromp curve. Figure 14 shows the results. In the process parameters (1#rotation speed of 220 rpm and an air inlet speed of 2.41 m/s), the cut sizes of the four structures are no vane-type 58.4 μm, direct-type 57.2 μm, L-type 55.6 μm and logarithmic-spiral-type 54.7 μm, respectively. The second process parameters are (2#rotation speed of 870 rpm and an air inlet speed of 2.83 m/s), the cut sizes of the four structures are no vane-type 19.8 μm, direct-type 17.2 μm, L-type 15.1 μm and logarithmic-spiral-type 14.5 μm, respectively. Therefore, the results indicate that, due to the decrease of the radial velocity in the annular region, the guide vane structures can effectively reduce the particle cut size.

Classification sharpness index is an important index which is widely used for evaluation of classification performance [28]. The employed calculation method of the index is K = $d_{75}/d_{25}$. Generally, the sharpness index is closer to 1, indicating better classification performance. According to the numerical simulation results, the classification sharpness index (K) and cut size are illustrated in Table 2. In the process parameters (1#rotation speed of 220 rpm and an air inlet speed of 2.41 m/s), the numerical simulation results indicate that the logarithmic-spiral-type guide vane structure reduced the cutting size by 6.3%, compared with other three results. The classification sharpness index (K) is closest to 1. Correspondingly, in the process parameters 2#, the cutting particle size is reduced by 23.7% and classification sharpness index (K) decreased from 3.05 to 2. In conclusion, the

numerical results indicate that the air guide vane can obviously improve the classification performance of the turbo air classifier.

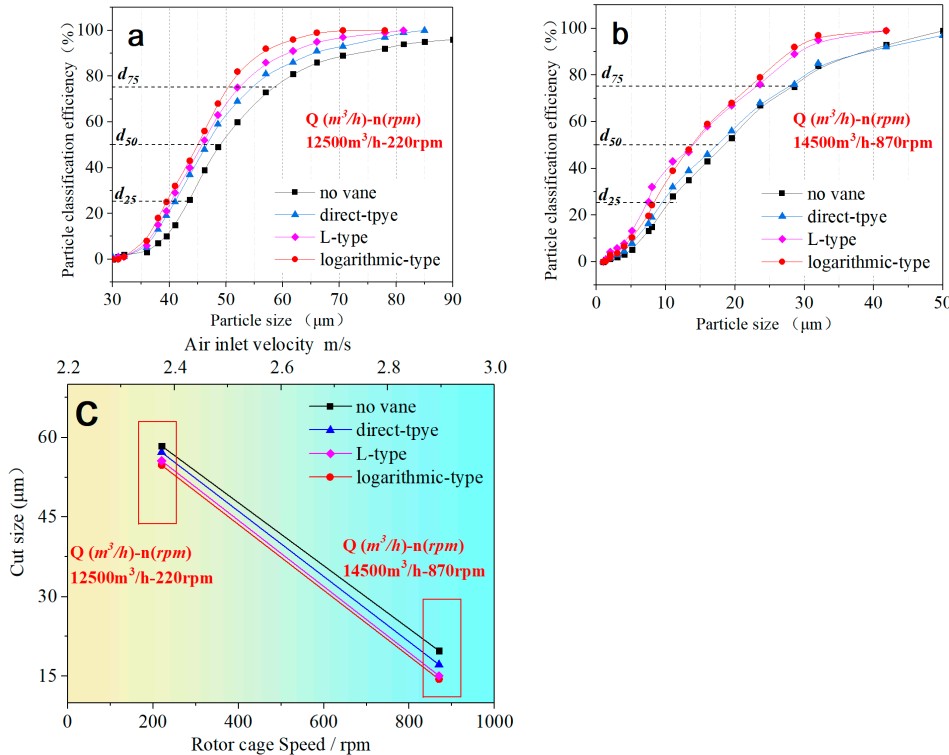

**Figure 14.** Numerical particle classification efficiency curves and cut size of four different guide vane structures by two processes. (**a**) Tromp curves by the processes of Air inlet flow rate: 12,500 m³/h and rotor cage speed 220 rpm; (**b**) Tromp curves by the processes of Air inlet flow rate: 14,500 m³/h and rotor cage speed 870 rpm; (**c**) Cut size by two different processes.

**Table 2.** The cut size and classification sharpness index for the different guide vane structures.

| v (m/s) | n (rpm) | Cut Size ($d_{50}$) μm | | | | Classification Sharpness Index (K) | | | |
|---------|---------|------------------------|------|------|------|-------------------------------------|--------|------|------|
|         |         | No Vane | Direct | L | Log | No Vane | Direct | L | Log |
| 3.41 | 220 | 58.4 | 57.2 | 55.6 | 54.7 | 1.65 | 1.59 | 1.54 | 1.52 |
| 3.83 | 870 | 19.8 | 17.2 | 15.1 | 14.5 | 3.05 | 3 | 2.64 | 2 |

## 5. No Guide Vane Experimental Results Verification

In order to further verify the reliability of numerical simulation, due to the limitation of industry expenses, material experiments are carried out with classifier equipment that has no guide vanes. The comparison between the experimental and the simulation results can provide a basis for the reliability of the other three simulation results. Equipment of the turbo air classifier is shown in Figure 15. The material experiments by two process parameters were carried out respectively. Figure 16a shows the experiment results of frequency percentage and cumulative distribution, according to the bypass values, which are obtained from two process parameters #1 and #2 are 11.3% and 2.6%, respectively. It was observed that the frequency percentage result of process parameters #1 are not very good. Consequently, as is shown in Figure 16c, in the partial classification efficiency (Tromp curve) of the process parameter #1 experiment, it can be found that there is an obvious fishhook phenomenon. The reason for this is that DPM models cannot predict this phenomenon due to the particle interactions being ignored in the numerical simulation. Therefore, the partial classification efficiency of the numerical simulation is better than the experiment.

However, comparison with the numerical and experimental results shows that the error of classification efficiency is very small. Therefore, it can be inferred that the results of numerical simulation have good guiding significance.

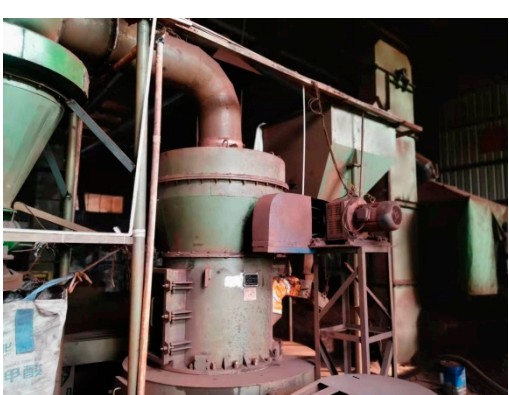

**Figure 15.** Photo of first stage turbo air classifier.

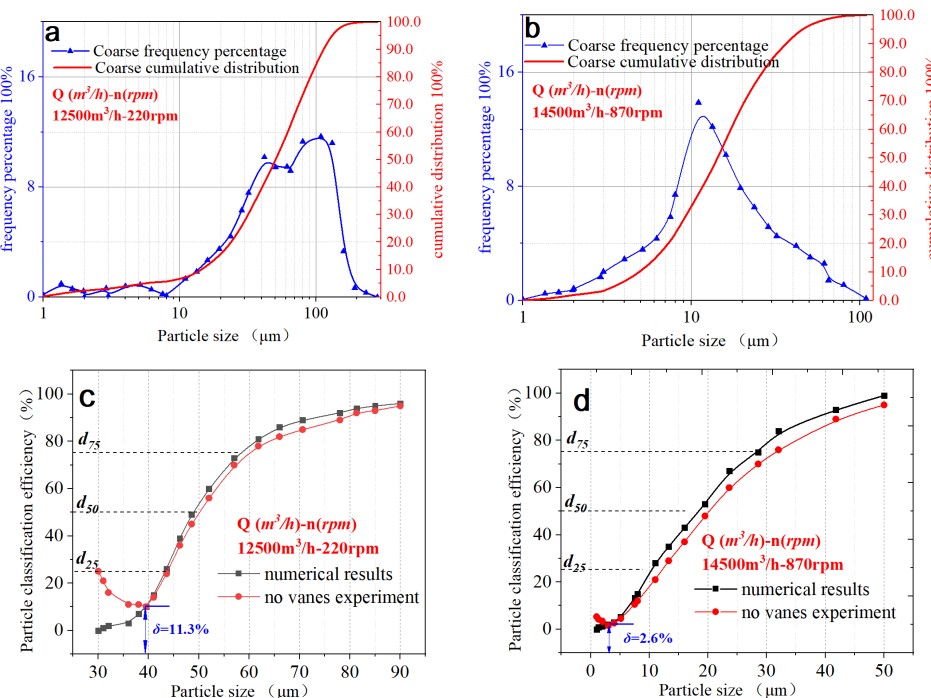

**Figure 16.** Experimental test results of two process parameters. (**a**,**b**) frequency percentage and cumulative distribution, (**c**,**d**) comparison of Tromp curves between the numerical simulation and experiment.

## 6. Conclusions

In the present study, three novel guide vane types of turbo air classifier were designed. Combined with no guide vane structure, the internal flow field of classifier is simulated by ANSYS-Fluent 20.0 in four cases. The effect of guide vanes on flow field and classification performance of the turbo air classifier was investigated. The following conclusions can be summarized below:

(1) The internal flow field stability of the classifier with air guide vane structure is obviously better than that of the classifier without the guide vane, especially in the annular functional region.

(2) The guide vanes can reduce the change of velocity gradient and vortex in the annular functional zone, improve its stability. By reducing the radial velocity of the annular

functional zone, the guide vane reduced the cutting size by 6.3% and 23.7% at two different process parameters.

(3) The guide vanes can reduce the turbulent dissipation rate at the inner edge of the rotor cage, and make the velocity distribution in the annular functional area more uniform.

(4) The logarithmic spiral guide vane is obviously better than other guide vane structures at improving the classification sharpness index (K). The numerical results of the two process parameters are 1.54 and 2 respectively. The L-shaped guide vane is also significant in improving classification performance.

**Author Contributions:** The author B.H. provided help in the preliminary investigation of this article. S.Z. provided resources such as experimental equipment. D.Q. Completed the numerical simulation. M.L. conducted in-depth research on the evaluation index of classification performance, then proposed the concept of the guide vanes. Y.Z. was incharge of the entire research experiment, statistics all the data and sorted it out. Finally wrote and revised the manuscript. All authors have read and agreed to the published version of the manuscript.

**Funding:** This research received no external funding.

**Data Availability Statement:** All data are presented in the article.

**Acknowledgments:** This project was supported financially by the Project of case Teaching for graduate degree of Yangtze University (No. YAL202106, Application of engineering software analysis). The National Natural Science Foundation of China (No. 52174018). CNPC Innovation Foundation 2020D-5007-0503. The authors would like to thank all the members of the project team for their support.

**Conflicts of Interest:** We declare that we have no financial and personal relationships with other people or organizations that can inappropriately influence our work, there is no professional or other personal interest of any nature or kind in any product, service and/or company that could be construed as influencing the position presented in, or the review of, the manuscript entitled, "Numerical and Experiment Investigation on Novel Guide Vane Structures of Turbo Air Classifier".

## Nomenclature

| | |
|---|---|
| $R_h$ | Radius of outer edge of rotor blade |
| $R_0$ | the distance from point on center line of guide vane to center of rotor cage |
| $V_{ht}$ | tangential velocity of airflow fluid element with radius $R_h$ |
| $V_t$ | tangential velocity of airflow fluid element with radius $R_0$. |
| $q_\varphi$ | flow rate |
| $h$ | the height of the rotor in blade classification region |
| $n$ | rotational speed of the rotor cage |
| $Q$ | Total volumetric flow rate of air |
| $d_p$ | Particle diameter |
| $d_{50}$ | Cut size |
| $u$ | the fluid phase velocity |
| $u_p$ | the particle velocity |
| $\mu$ | kinematic viscosity of fluids |
| $\rho$ | fluid density |
| $\rho_p$ | the particle density |
| $Re$ | the relative Reynolds number (particle Reynolds number) |
| $C_D$ | drag coefficient |
| $F_x$ | additional acceleration (force/unit particle mass) term |
| $F_D(u - u_p)$ | the drag force per unit particle mass |
| $\varphi$ | direction angle |

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
