# Peer review of "Numerical and Experiment Investigation on Novel Guide Vane Structures of Turbo Air Classifier"

_processes, doi:10.3390/pr10050844_

Round 1
Reviewer 1 Report
Numerical and Experiment investigation on novel guide vane structures of turbo air classifier
The authors investigated the influences of the guide vane structures on the flow field and classification efficiency of a turbo air classifier. The manuscript is well-written. The reviewer found the statements, the flow of the manuscript and the discussions sufficient. Minor revision addressing the below mentioned points is recommended.
Please revise the flow of the first few sentences. I would expect to hear directly about powder preparation and its importance but the term drilling appeared all of a sudden.
“Drilling and completion fluid and fracturing fluid play a decisive role in the production efficiency of unconventional oil and gas fields. Therefore, many researchers have conducted lucubrate studies on the formulation of drilling and completion fluid and fracturing fluid. With the development of research, the particle size distribution of solid particles in drilling and completion fluid and fracturing fluid formula has an important impact on its efficacy. Practical studies have verified drilling fluid and completion fluid with iron ore powder has significant effect, which is a comparatively effective new technology.”
Section 3.1, please revise the statement as “cross section is roughly simplified as illustrated in Fig. 5”
Section 3.2: “Combined with the specific structural parameters of the classifier in this paper (classifier runner blade parameters: length 50 mm, width 4 mm, height 480 mm. inlet mounting angle 60°) and process parameters.” The reviewer did not understand what is mentioned. Please be more clear in the statement.
The reviewer suggests putting the linear velocity information into Section 4.1.1. What corresponds to 220rpm should be mentioned.
Section 4.1.1. What do you mean with “more stable air flow”. How did you come up with such a conclusion? What does stability mean in a turbo air classifier? No vane condition also seems to have the stability if I catch your thoughts correctly. Please be more explicit on the explanations given in Fig. 6.
Section 4.2.2. Please change the heading as “Comparison of the turbulent….”
The conclusion you drew in Fig.11a is really important (the case with logarithmic type)
Section 4.3 is delightful. Please also indicate the bypass values. It seems the logarithmic type has the least amount of bypass portion and in the meantime has less cutsize and higher sharpness values compared to other structures. These outcomes are valuable. Please express it in this way to further emphasize the importance and the quality of your work.
Author Response
"Please see the attachment."

Reviewer 2 Report
The authors compare three types of air guide vanes including direct-type, L-type, and logarithmic spiral type, using both numerical simulations and experiments. This work could be interesting to the community if some improvements were made. I have several comments on this work.
- Section 3.3 lacks proper citations for those equations. Equation 7 has a sourcing term Si. What is the meaning of this term in this work? Equations 8-12 require more detailed descriptions of the meaning of those variables.
- It would be clearer if the authors added the corresponding type names on each subfigure in Figures 8, 9, and 10.
- No guide vane experimental results are used to verify numerical simulations. This is not an appropriate verification since this work mainly compares differences among different air guide vanes. This work would be much stronger and better if experiments with air guide vanes could be performed to verify simulation results.
Round 2
Reviewer 2 Report
The authors have addressed my comments properly.